# Food Identities, Biocultural Knowledge and Gender Differences in the Protected Area “Sierra Grande de Hornachos” (Extremadura, Spain)

**DOI:** 10.3390/ijerph17072283

**Published:** 2020-03-28

**Authors:** Lorena Gutiérrez-García, Juana Labrador-Moreno, José Blanco-Salas, Francisco Javier Monago-Lozano, Trinidad Ruiz-Téllez

**Affiliations:** 1Department of Plant Biology, Ecology and Earth Sciences, University of Extremadura, 06006 Badajoz, Spain; lorenagg@unex.es (L.G.-G.); labrador@unex.es (J.L.-M.); truiz@unex.es (T.R.-T.); 2Department of Business Management and Sociology, University of Extremadura, 17071 Cáceres, Spain; fjmonagolo@unex.es

**Keywords:** culinary identity, food, ethnobotany, agroecology, sustainable development, economic growth, cultural wealth

## Abstract

A food tradition not only corresponds to the vital need to be nourished every day, but is part of the particularity of a territory as a consequence of its history, traditions, natural heritage, and capacity for ecological and social resilience. In the search for culinary identity, a valorization of a rural territory of high identity potential is carried out, such as in the environmental protection area “Sierra Grande de Hornachos” (Extremadura, Spain), and specifically the town of Hornachos. For this purpose, a series of workshops and interviews were held for men and women who had lived most of their lives in Hornachos and who were older than 70. Information on the food uses of wild and cultivated plants, as determined by the Cultural Significance Index (CSI) for 79 species, was extracted from the interpretation of the data collected. In addition, new uses were collected in Extremadura for 16 plants and in Spain for 3, with some of these data being of particular significance in the culinary culture of Hornachega. We conclude that the area “Sierra Grande de Hornachos” forms an environment of great culinary identity that must be preserved, not only for its heritage interests but also for its agroecological ones, which could be translated into measures of wealth creation and development.

## 1. Introduction

In building the identity of a country, a region, or a community, food is the central axis of human relations, and also presents itself as a carrier of diverse meanings which acquire value from the natural and cultural history of the area’s inhabitants.

Food is an important element that supports social groups to become aware of their differences and ethnicity—understood as the feeling of being part of a different cultural entity—so that sharing it can mean recognition and acceptance/incorporation of these differences [1].

To this same extent, the kitchen, considered in its broadest sense, in which not only the place for culinary procedures but also for the practice of sociocultural values, and is a reflection of the society it is situated in since each social group has a table of references that guides the choice of foods that confer a particularity, differential, distinctive experience [2].

In this context, food identity is the expression of the cultural, economic, and agro-environmental particularity of a territory and the people living there. It is the consequence of its history, its traditions, the interactions between the different civilizations that have occurred, the richness of its biocultural heritage and the knowledge of its management, and capacity for ecological and social resilience as a community. It also has an important identity role, as it fosters a sense of belonging to a community with its own identity.

Agri-food systems are the processes and infrastructure involved in feeding a population. The sustainability of agri-food systems can be approached by two paradigms the sustainable development (focused on agriculture and the environment) and the relocalization paradigm. The latter very used by sociologists and mainly based on alternative food systems [3]. Today, there is a loss of food culture due to globalization and market demands, causing the proliferation of non-nutritious meals and disrespect to the natural environment [4]. We are less and less close to what we eat [5], and this deterioration in diet is affecting even our health, with the rise and development of chronic diseases [6] An approach dealing with agriculture, food, and environment should consider the diversities of potential and initiatives in each specific territory [3]. The first steps should include an ethnographic assessment that allows the identification of traditional knowledge [7]. We must rescue and revalue that wisdom, remaining aware of its potential to address the loss of identity and resources. To maintain our people, promote sustainable development, and maintain the balance of our society, a new bond between humans and their natural environment and profound changes in the symbolic assessment of food—not just the act of nutrition, but the production, processing, and marketing of food—are required [8].

The new food modernity demands safe and quality food, incorporating specific values associated with a territory, a nature, a culture, and a specific form of production and processing, and from within the rural context, this modernity alternative is also generating broad sectors interested in sustainable development and the recreation, from various spheres, of a culinary identity that links a territory and a culture with the elaboration and consumption of a type of food [9].

In 2006, food culture was recognized by UNESCO as a value worthy of patrimonialization; that is why it is now considered as an intangible heritage of cultures, and preserving the culinary traditions of different countries has become of great interest, not only to safeguard their biocultural memory but as a major economic resource for sustainable development [10] and a multifunctional contribution to the well-being of the community and the protection of its natural resources.

The construction of food identity at different levels and its relationship with a territory is an issue that arouses enormous interest not only for science but also for policy, economy and culture managers, as it is “huge the diversity of practical applications that could detach from it (cultural heritage, health, tourism, economic development, food marketing, territory management, legal protectionism of some products, etc.)” [11].

Traditionally, the rural world is considered the source of traditional culinary practices—the place where the original knowledge and customs reside. This environment was always multifunctional and self-sufficient, a place where production, processing, and consumption was carried out in a manner respectful to food and with the supplying nature of food, “and although this was done in an intrinsic way in most cases, the settlers created a fairly strong identity and this was transmitted from generation to generation, creating an invaluable cultural heritage” [12].

The strong identity and culinary variety of a region represent one of the main strategic elements of the local gastronomy. Extremaduran cuisine still maintains a strong relationship with the territory in which it develops, and this can be used as a development resource, especially for the rural environment. Extremadura has a great abundance and quality of local products, and this richness is made even greater when we talk about gastronomy associated with the use of wild products and cultivated varieties typical of a locality or region, and that constitutes a cultural richness that, as mentioned above, is in danger of extinction [13].

The culinary identity, as well as the specificities and potentialities of rural territories through the rescue of the food cultural tradition and the value of traditional biocultural knowledge about natural and cultivated biodiversity within a specific territorial area, is proposed as a topic of study. We focus on the protected area “Sierra Grande de Hornachos” (Extremadura, Spain) a territory where there is a great plant diversity, where it is estimated that there may be more than 830 plants with known utilities for humans in the Spanish territory, uses that are determined by the culture and peculiarities of each place [14]. Hornachos owns strong agglutinates of identity—sources that are cultural, natural, historical, and geographical—and utilize an agroecological approach as a possible but not unique catalyst conceptual framework to energize the different dimensions of sustainable endogenous rural development, this makes it a place where there are determining factors for the existence of a wealth of culinary knowledge, associated with its natural environment.

We will also take into account the familiarities of men and women, as previous studies show that this is a differentiating factor in terms of the identity traits of a place [15] because, traditionally, the roles of each of the genres have been different. Women have always been associated with the field of home and childcare and parenting, while men spent less time in the house and used to perform jobs associated with agriculture and livestock maintaining a more direct relationship with the natural environment.

This work addresses two basic questions at two different times, taken as a reference to the ethnobotanical method described by Rodríguez Guerra and co-workers [16], who speaks of an implementation phase through surveys, although in this case it will be done through workshops and a generalization phase in which an interview is included. Firstly, we intended to obtain precise information on the traditional culinary knowledge of plants already referenced in other publications and secondly, in the case of a significant traditional gastronomic richness in the area, we intended to deepen the concept of culinary identity and to locate food plants with a degree of significance whose use could be particular to or identifying of the town of Hornachos.

In order to respond to these objectives, the development of a quantitative/qualitative mixed methodology was considered relevant, allowing a better understanding of the object of study [17]. Two types of instruments or techniques were used for data collection: workshops and interviews.

Quantitative logic prevails in this research by using workshops that collected the data via a structured questionnaire for a group of men and women who were shown, separately, a series of photographs of plants growing in in order to obtain information on the food uses of these species while discerning possible gender differences.

In addition, the data were analyzed qualitatively following collection through interviews on open-type issues, addressed to a narrower group of people than in the workshops, in order to obtain more accurate information on some of the species of interest and the gastronomic culture associated with them.

## 2. Materials and Methods

### 2.1. The Study Area: Sierra Grande de Hornachos

The town of Hornachos (Badajoz), a town belonging to the province of Badajoz (Extremadura) southwest of the Iberian Peninsula and located on the edge of the protected area “Sierra Grande de Hornachos”, was selected as the study area. It is a town located on the south face of Sierra Grande, a place that, due to its plant and faunistic wealth, has various types of protected space [18], from the regional (Regional Zone of Interest (RZI)) to the international (Site of Community Importance (SIC), Special Protection Area for Birds (SPA), and Special Area of Conservation (SAC)), as well as being included in the Biodiversity Conservation Network of the European Union (Figure 1).

This mountainous set stands between wide expanses of plains, which makes it a redoubt of the Mediterranean forest in the heart of Lower Extremadura, and it shapes the divide between the regions of Tierra de Barros (to which it belongs), Campiña Sur, and La Serena.

At the climatic level, it is within the subtype of Mediterranean Atlantic-Continental climate of semi-arid type, characterized by above-average temperatures and very low rainfall.

These climatic and geographical characteristics allow the existence of an abundant fauna (with more than 220 vertebrates) and plant diversity, which, according to recent surveys, could exceed 1300 species [14].

The economy of the municipality revolves around the use of natural resources and the agricultural activity of dryland (vine, olive, and cereal), being minority livestock.

In addition, the territory where this natural space is integrated presents an enormous cultural richness due to the coexistence of Muslim, Jewish, and Christian cultures.

The presence of interculturality persists in the gastronomy of the inhabitants of Hornachos, fundamentally that of the so-called Moors, of which the Moorish orchards are still preserved, and where planting techniques and knowledge have endured, as well as native varieties of enormous genetic, cultural, and commercial value [14].

This is, therefore, a territory with a rich cultural and agro-environmental heritage; however, this complexity has not been expressed in opportunities for the dynamization of the local economy and the creation of bonds of a union in the community. Instead, it is returning back to an old approach that prioritizes conservation of the natural environment and agricultural production; currently, this is clearly observed in the social dynamics.

In these contexts, where rural development does not imply progress, and in areas where environmental protection of the territory is a brake on development, agroecology as an integrative science can provide a new paradigm and a holistic approach, which recognizes the complexity of the rural world and builds endogenous development from multiple dimensions, namely, ecological, social, economic, political, and technological, thus transcending fragmented and short-term vision.

### 2.2. Plant Material

As a working tool for the workshops, we used the guide “Plants of the Sierra de Hornachos (modified and expanded)” [19]. This book includes a total of 337 plant taxa that can be found in the protected area “Sierra Grande de Hornachos”. This guide is presented as a tool for the visual identification of plants as tokens that include several photographs of each species at different stages of its vegetative cycle, as well as a brief description. Its easy handling meant it was considered the best option to achieve the proposed goal. Of the 337 plant taxa, 79 listed in the Spanish Inventory of Traditional Knowledge relating to Biodiversity [20] were selected, which have a proven use as “Food”, as follows:

*Allium ampeloprasum* L. 1; *Allium neapolitanum* Cirillo 2; *Andryala integrifolia* L. 3; *Arbutus unedo* L. 4; *Arisarum simorrhinum* Durieu 5; *Arum italicum* MilL. 6; *Arundo donax* L. 7; *Asphodelus albus* MilL. 8; *Bellis perennis* L. 9; *Bituminaria bituminosa* (L.) C.H. Stirt. 10; *Borago officinalis* L. 11; *Bryonia dioica* Jacq. 12; *Calamintha nepeta* (L.) Savi 13; *Calendula arvensis* (VailL.) L. 14; *Campanula rapunculus* L. 15; *Capsella bursa-pastoris* (L.) Medik 16; *Carthamus lanatus* L. 17; *Ceterach officinarum* WilL. 18; *Chenopodium murale* L. 19; *Chenopodium* sp. 20; *Chondrilla juncea* L. 21; *Cichorium intybus* L. 22; *Cistus albidus* L. 23; *Cistus ladanifer* L. 24; *Crataegus monogyna* Jacq. 25; *Cynara humulis* L. 26; *Cytinus hypocistis* (L.) L. 27; *Digitalis thapsi* L. 28; *Dittrichia viscosa* (L.) Greuter 29; *Foeniculum vulgare* MilL. 30; *Galactites tomentosus* Moench. 32; *Helichrysum stoechas* (L.) Moench 33; *Hypericum perforatum* L. 34; *Lonicera etrusca* Santi 36; *Lonicera implexa* Aiton 37; *Malva sylvestris* L. 38; *Mantisalca salmantica* (L.) Briq. and CavilL. 39; *Marrubium vulgare* L. 40; *Medicago orbicularis* (L.) BartaL. 41; *Mentha pulegium* L. 42; *Myrtus communis* L. 43; *Olea europaea* subsp. *europaea* var. *sylvestris* (MilL.) Lehr. 44; *Onobrychis humilis* (L.) G. López 45; *Origanum vulgare* subsp. *virens* (Hoffmanns. and Link) Bonnier and Layens 46; *Oxalis corniculata* L. 47; *Papaver rhoeas* L. 48; *Phillyrea angustifolia* L. 49; *Phlomis lychnitis* L. 50; *Pistacia lentiscus* L. 52; *Portulaca oleracea* L. 53; *Quercus rotundifolia Lam.* 54; *Quercus suber* L. 55; *Reichardia intermedia* (Sch. Bip.) Samp. 56; *Retama sphaerocarpa* (L.) Boiss. 57; *Rosmarinus officinalis* L. 58; *Rubus ulmifolius* Schott 59; *Ruscus aculeatus* L. 60; *Scandix pecten-veneris* L. 61; *Scirpoides holoschoenus* (L.) Soják 62; *Sedum brevifolium* DC. 63; *Silene vulgaris* (Moench) Garcke. 64; *Silybum marianum* (L.) Gaertn. 65; *Smilax aspera* L. 66; *Solanum nigrum* L. 67; *Sonchus oleraceus* L. 68; *Tamus communis* L. 69; *Thapsia villosa* L. 70; *Thymus mastichina* (L.) L. 71; *Ulmus minor* MilL. 72; *Umbilicus gaditanus* Boiss. 73; *Umbilicus rupestris* (Salisb.) Dandy 74; *Urginea maritima* (L.) Baker 75; *Urtica urens* L. 76; *Veronica anagallis-aquatica* L. 77; *Viburnum tinus* L. 78. And cultivated species (+) *Fragaria vesca* L. +31; *Laurus nobilis* L. +35; *Pinus pinea* L. +51; *Vinca difformis* Pourr. +79.

With these plants, a collection of 79 photographs was prepared in a PowerPoint presentation, which was used for the development of the first part of the fieldwork: the workshops. In these images, instead of the names, only the reference number that is observed in the table above appeared and was accompanied by the symbol “+” for those that corresponded to cultivated species.

In the case of interviews, no photographs or other visual supports were used. It was established that the issues would be aimed at obtaining more data on those species in the list of 79 taxa (without dismissing other non-listed ones that might have been of particular interest), which, after the workshops, were pointed out by the participants for their food uses, dismissing those that did not have any culinary uses.

### 2.3. Design of Workshops and Interviews

#### 2.3.1. Selection of Participants and Participation in Workshops

The population of Hornachos, according to data from the National Statistical Institute [21], is approximately 3639 inhabitants, of which the population over 70 years of age is composed of some 291 men and about 355 women [22]. Considering that these data may vary slightly from the current figures, a group of 31 men and 39 women was chosen as the sample, representing around 10% of the inhabitants of this locality, which was considered a representative and viable number for this study.

All of them, prior to the workshops, had to complete and sign an informed consent form on the research and processing of the data they were going to provide.

The workshops took place in a friendly environment, which is essential for this type of methodology for getting participants to collaborate and to ensure the data would be reliable.

Men and women were summoned on different dates, to prevent them from interacting. The interviews lasted 10 days, conducting one workshop per day (five for each gender), with groups of 7–8 women and 6–7 men.

The activities consisted of showing each group the photographs of the 79 plant species (the same in all workshops) for participants to comment on what they knew about each of them. The answers were recorded by the interviewer on model cards, such as the one in Table 1.

For the design of the data collection sheet, the 12 subcategories of uses in human–plant feed, described in the Spanish Inventory of Traditional Knowledge relating to Biodiversity [20], were considered. Each of these subcategories was quoted verbally during the workshops for each of the plants, while the interviewer wrote down the information given by the interviewees. These subcategories refer to food types and their nutritional importance. Each of them (namely, Uses 1 to 12) is detailed below, including a brief description:

(1) Vegetables: flowering, leafy, stem, whole-plant, and fruit vegetables. Pods and seeds of immature legumes are included, as well as flowers used in salads and gills. (2) Roots, bulbs, tubers and rhizomes: rhizomes of grass, enea, some ferns, and their derivatives. (3) Sweet fruits: fleshy fruits rich in sugars that are normally consumed as dessert, and their derivatives. Includes dried fleshy fruits. (4) Dry and oleaginosus fruits: high-fat nuts. Seeds are included. Edible oil fruits. (5) Cereals and pseudocereals: fruits rich in carbohydrates such as wild cereals and pseudocereals and their derivatives. (6) Fat: oils, fats, and butters. (7) Alcoholic drinks: fermented, distilled, and macerated. (8) Non-alcoholic drinks: water. Coffee or chocolate substitutes. Food infusions. Musts, juices, and other refreshing beverages. (9) Condiments: substances used to give flavor, smell, and color to meals, often with preservative intent. Includes aromatic herbs. Unintentional seasoning preservatives. (10) Sugars and sweeteners: sugar plants. Other plant-based sweeteners. (11) Candies and chewing: direct consumption (in the field) of different vegetable parts that are sucked or chewed to cool down, to remove hunger, as entertainment, or for the pleasant taste. Candy, gum, and marshmallow. (12) Other food uses: water purification, identifying mushroom toxicity, counteracting the bitterness of other food. Yeast.

A total of 3 h was dedicated to each workshop, of which 20 min was rest time, so the average amount of time used for each plant was 2 min.

Finally, the data were taken to the Department of Plant Biology of the University of Extremadura, where they were digitized.

#### 2.3.2. Selection of Participants and Participation in Interviews

The interview is a valuable technique for qualitative research, as it allows the researcher to collect detailed information on a specific topic shared orally between informant and interviewer. An unstructured direct interview was chosen as no categorization was intended to be imposed a *priori* on the responses, which could limit the field [23].

We selected four people (two men and two women) aged over 70, all of them from Hornachos and who had lived most of their lives in this town, who were shortlisted during the course of the workshops for their high degree of knowledge of the culinary uses of plants in the Sierra Grande space.

As in the workshops, prior to conducting the interviews, participants had to sign their informed consent on the processing of the data they were going to provide.

The interviews were conducted during the month of January 2020, individually in the afternoon and on different days, adapted to the availability of the interviewees.

A number of open-type issues were designed to obtain information on those plant species most significant to the Hornachega population from the point of view of their culinary use, with the aim of identifying peculiar uses or identifying people of the Sierra Grande protected area.

The collection of the data was done by means of annotations in field notebooks with the help of a table (Table 2) and recordings in video format, which were subsequently analyzed for the extraction of the information.

Interviews were aimed at achieving three objectives or variables: (1) Importance of wild plants and local varieties in the daily life of the Hornachega population. (2) Most relevant species, for their abundant use or particularity. (3) Uses and elaborations associated with each species cited. Thus, it was intended that, despite open conversations, the interviewer would redirect the informant from a general level to answering specific questions [24]. The total number of issues was six, with two issues for each variable.

### 2.4. Data Digitalization

#### 2.4.1. Workshop Data Digitalization

The data collected in the tabs of each of the plants were entered in an Excel spreadsheet, such as the one shown in Figure 2. The rows contain the 79 scientific names in two blocks of repeated columns with numbers from 1–12, which correspond to the 12 subcategories of food types considered. The cells in the left block contain the data obtained from the men, and the cells of the block on the right contain those obtained from the women.

The definition of each row is then provided according to the textual description of [15]:“Row n: Proportion of workshop participants who confirmed that use category. It is expressed as per participant, not as a percentage. Minimum value 0. Maximum value 1 (see Figure 3 for transcription of the example of Table 1).Row i (management): Significance of the plant from an agronomical point of view. For wild plants, value = 1. For cultivated plants, value = 2.Row e (preference): Proportion of the workshop participants who selected that use category as “the most preferred”. It is expressed as per participant, not as a percentage. The maximum value 2 was given to the preferred category. Non-preferred categories, value = 1. Non-used categories = 0.Row c (frequency): Proportion of the workshop participants who selected that use category as “the most frequently used”. It is expressed as per participant, not as a percentage. The maximum value 2 was given to the selected category. Non-selected categories, value = 1.Row (i*c*e): Value obtained for a category of use of the species in the workshop. Maximum value (2 × 2 × 2) = 8. Minimum value (1 × 1 × 1) = 1. Non-used categories = 0.Column ∑n: Minimum value = 0. Maximum value = 12. H = the highest value obtained in the workshops by one species.Column ∑(i*e*c): Global value of all the uses of the species. The maximum value that one species can obtain is (2 × 2 × 2) + [11 × (2 × 1 × 1)] = 30.”

#### 2.4.2. Interview Data Digitalization

The information collected in each of the interviews was collected in a synthesized Excel table that included the parameters Variables, Questions, and Informants (Figure 4).

### 2.5. Analysis of Data

#### 2.5.1. Quantitative Analysis of Data: Cultural Significance Indexes (CSI), and Calculations and Statistics

The cultural significance indexes (CSI) mathematical function [25] modified by [15] was used for this study as a result of a critical review of the applicable indices [26].

The CSI assesses the relevance that a species can have for an informant among a set of species. In its original design, it is not necessary to have a pre-established classification of uses. The informants can classify them as they decide during the interview.

Our proposal, apart from considering a closed list of 12 types of food uses of the Spanish Inventory of Traditional Knowledge relating to Biodiversity [20], calculates the value of the H parameter, following its previously mentioned definition. The mathematical function to calculate the modified CSI is as follows:
CSI = CF*[∑(i*e*c)_i_] = [∑n/H]*[∑(i*e*c)_i_]
(1)
where *n* = use, *i* = management, *e* = preference, *c* = frecuency, and correcting factor (CF) = ∑n/H.

In theory, any species might reach the maximum variety of uses, which was 12 in our classification system. However, it is well known that, in practice, the maximum number of uses of one species will be lower. The number depends on the plant and on the knowledge of the participants of a concrete workshop. In order to compare the results obtained in different workshops, we employed the CF, as defined above. The CF modulates the knowledge about the uses of a species in the context of the proper workshop. It is very useful when we need to measure or compare between species or group of species, and informants or groups of informants. The CF maximum value = 1. The CF measures the degree of deviation of the uses of a species from the most useful situation found in that workshop, so it allows interpretation of the results of the questionnaire in the context of the workshop itself. For this reason, it is a very important parameter to distinguish the knowledge of two replicated groups: the group of women and the group of men. It is an intra-quantitative comparative parameter.

The CSI is the total sum of different types of uses of the plant expressed by numbers. It helps to order lists of species upon the importance of their utility and to compare them. It considers not just the frequency of use but also a subjective valorization of quality, indirectly measured through the parameter *e* (=preferred category). Some authors place much weight on this formula because it is considered that knowing or ignoring a wild plant has a very different cultural, rather than agroforestry, explanation. This appreciation is influenced by the former definition of CSI from Da Silva et al. [25].

Individual CSI values of the 79 plants from the perspective of the 12 categories of food uses were obtained. Knowledge from men was quantified as the CSI men summary and abbreviated as CSIm. The women’s knowledge, abbreviated as CSIw, was calculated the same way.

Non-parametric statistical tests were applied to compare the groups of CSI values (Wilcoxon test, with SPSS V.20 for Windows, IBM, New York, NY, USA).

#### 2.5.2. Qualitative Analysis of Data

For the qualitative interpretation of the data collected during the interviews, they were extracted with the help of Excel tables considering species and their food uses, taking as reference the 12 subcategories of use in food described in the Spanish Inventory of Traditional Knowledge relating to Biological Diversity [15] and elaborations cited by the interviewees, classified according to the culinary categories proposed by Bertrand [27]. Subsequently, a critical analysis of the same was carried out.

## 3. Results

### 3.1. Workshop Results

Below are the uses that the different species of plants proposed in this study were found to have. The results obtained for the workshops of men and for the women’s workshops are listed, and a comparison is made between the two groups (row results are in the Appendix A).

Considering the data collected as a whole, a total of 63 uses were recognized for 36 plants. Men recognized 47 uses for 31 species, while women associated 55 uses with 33 species.

The largest number of usage categories assigned to a plant (H) was five. In order to facilitate data comparison and reduce possible errors, all values obtained for CSI were multiplied by 1000 (Table 3, Table 4 and Table 5). The values of CSIm*1000 and CSIw*1000 for total taxa (*n* = 79) were then compared using the Wilcoxon test, and no significant differences were found (*p* > 0.001).

### 3.2. Interview Results

The following tables and figures show the plants considered by the interviewees as the most important in the culinary culture of Hornachos (Table 6), as well as the most common uses and elaborations. For the 12 food uses, 7 uses were mentioned during the interviews: Vegetables (Use 1 from the list), sweet fruits (Use 3), dry and oleaginous fruits (Use 4), alcoholic drinks (Use 7), non-alcoholic drinks (Use 8), condiments (Use 9), and candies and chewing (Use 11).

On the other hand, 13 elaborations (Figure 5) were the most common for the 31 species obtained, which in turn have been grouped into four types of dishes: first courses, sauces, confectionery, and drinks [27].

## 4. Discussion

### 4.1. Species

Hornachos is a small municipality, eminently agricultural and livestock, which, like many others in its environment, suffers a constant loss of population due to migration to higher density nuclei in search of work [28].

In the fight against depopulation and the promotion of new development strategies in rural areas, agroecology plays an important role. It is defined as a new field of knowledge that integrates the knowledge and experiences of traditional agriculture, coupled with a strong ethical component, in order to place food at the center of the life of communities as an identity factor [29].

To promote changes and hopeful premises from agroecology, it is necessary to know some of the more or less paradoxical features of our food at present. During the development of workshops and interviews, the effect that industrialization and new ways of life are having on the gastronomic heritage was evident. These lead to the loss of family farming and the heterogeneity of culinary knowledge since, in most cases, there was talk of past customs that are already obsolete—of something that was done, but that no longer happens.

Another aspect that the participants themselves perceived was that there was not great knowledge of plant species in their locality; no ingrained and remarkable tradition of culinary exploitation of their wild species and cultivated land varieties. The results reveal that this idea is not entirely true, but that it is on its way to being true. There is still a fairly high recognition of plants since, of the 79 species proposed in this work, 66 were recognized, and, of them, some use was associated with 36. It is true that the loss of ethnobotanical culture is evident in the type of responses obtained since, sometimes, the participants knew that there was a certain use, but they had not used the plant themselves and could not explain anything else about that use.

In analyzing the results, it was noted that, during the interviews, a total of 31 wild and cultivated plant species were cited, of which 11 were part of the list of 79 taxa proposed in this work for the realization of the workshops. This number is very significant since, in this part of the fieldwork, only four cultured species were addressed, but in interviews, due to their openness, the number of cultivated species mentioned increased, which may be due to their use being directly associated with human food.

If we compare the 11 plants obtained in the interviews included in the list of 79 proposals for the workshops, it is noted that six of them (4. *Arbutus unedo* L., 25. *Crategus monogyna* Jacq., 30. *Foeniculum vulgare* Mill., 46. *Origanum vulgare* subsp. *virens* (Hoffmanns. and Link) Bonnier and Layers, 54. *Quercus rotundifolia* Lam., and 59. *Rubus ulmifolios* Schott) are among the top 11 best rated in the overall workshops, while the other five are among the 29 with a higher total CSI. Interestingly, for both men and women, the first 11 species were the same as in global analysis, with small variations in order.

Although the interviewer guided the various conversations according to the objectives, 10 of the species (*Asparagus albus* L., *Asparagus acutifolius* L., *Scolymus hispanicus* L., *Rumex pulcher* L., *Rorippa nasturtium-aquaticum* (L.) Hayek, *Quercus suber* L., *Prunus domestica* L., *Phlomis lychnitis* L., *Crataegus monogyna* Jacq., and *Citrus × sinensis* (L.) Osbeck), were mentioned by more than 50% of the interviewees, showing that these are well-known, everyday species, many of which are still included in the local gastronomy.

In this way, we obtained a sample of 16 plants that we can consider highly valued in the gastronomic culture of Hornachos. Of them, *Crataegus monogyna* Jacq. is the one that obtained the best result after analyzing all the data collected in this work.

### 4.2. Uses

Note that most species were found to have one use (16 species), followed by two uses (13 species), and five uses was the maximum associated with the same plant (2 species).

As for the number of plants per category of use, three of the uses were not associated with any of the taxa (cereals and pseudocereals (Use 5 from the list), sugars and sweeteners (Use 10), other food uses (Use 12)). Of the remaining nine, the ones that appeared the most were non-alcoholic drinks (Use 8), associated with 57% of plants used in Hornachos, and vegetables (Use 1) for 48% of plants used in Hornachos. This is because many of the species have medicinal uses as infusions and tesanas. Because the consumption of these non-alcoholic beverages may not always be associated with a medical condition, even if they have a healing property, this is considered a food use. The seven uses obtained in interviews, although not quantifiable, again do not include Uses 5, 10 and 12, so these are not relevant in the culinary tradition of “Sierra Grande”.

From another perspective, it was observed that the role of the species under study in the diet as vegetables in the production of stews, omelettes, and other recipes is very important in Hornachos.

On the other hand, in the search for one or more distinctive factors that can define the culinary identity of the protected area “Sierra Grande” and therefore the town of Hornachos, the results extracted in this work have been contrasted with those collected in the Spanish Inventory of Traditional Knowledge relating to Biodiversity [20], such that 16 plants obtained food uses in Hornachos, of which no such information was available for the Extremaduran autonomous community.

In addition, by expanding the search range, it was seen that three species (26. *Cynara humulis* L., 55. *Quercus suber* L., and 75. *Urginea maritima* (L.) Baker) had a total of three uses (in the established food categories) that do not appear in the Spanish Traditional Knowledge Inventory, for any region of Spain (Table 7).

These uses could constitute a hallmark of Hornachos and the protected area of “Sierra Grande”; however, they are not widespread in the population. In the case of *Quercus suber* L., a new use not registered in Spain was found, but that was only mentioned by one person during workshops and interviews. However, during individual interviews, gender trees appeared for *Citrus* and *Pyrus*, with local varieties grown, and these cases can be considered Hornachos identity cards.

### 4.3. Gender Differences

Analysis of the results of the workshops did not reveal a difference in knowledge based on gender. This was also observed in the results obtained in the second part of this investigation.

This result is explained in the fact of the culinary identity being the self-representation of a place [30], dependent on its history and the customs and traditions that occur there. In the case at issue, there is no greater importance of the plants under study for men than for women when we analyze them globally, but, in turn, there are certain species that stand out in one group or another.

Currently, the rural world is undergoing a continuous process of change in occupations and ways of life that is having an impact on the division of tasks by gender [31], but it is known that, traditionally, women have been linked to reproductive functions and the performance of tasks within the domestic field, while men worked in the field. This allocation of tasks also occurs and has occurred in Hornachos, but the fact that, from a gender perspective, there was not greater knowledge on the part of either men or women is probably because, although their roles are different, in this case, there is a confluence of knowledge since men were usually the ones who collected the plants and later brought them home, while women were responsible for including them in dishes, which required both the species and their food uses to be known.

## 5. Conclusions

It was shown in this study that the cultural richness associated with the traditional use and consumption of wild and cultivated plants of the protected area “Sierra Grande de Hornachos” constitutes a culinary identity of great interest for its conservation. Some differences have been obtained in terms of the species most valued by men and women and the most relevant uses for each of these groups, but they are not significant, possibly because food is something relative to the whole family, regardless of the role played by each actor; all actors know of the plants that are part of their diet and their elaborations. Three possible new uses for certain plants were also identified, making it even more interesting.

In today’s quest for exclusivity and innovative ideas that inspire measures for development and progress, mainly in rural areas, Hornachos presents itself as a high-quality place in its traditional gastronomy, which could become a resource for agricultural tourism and sustainable agriculture.

Traditional local knowledge could become a powerful way to solve the problems facing the rural world. Recent studies and initiatives aimed at the reintroduction of wild plants and local varieties of species highlight the potential of traditional foods as an alternative to industrialized production, showing the emergence of a new model of natural resource management, based on local knowledge and agronomic innovation with agroecological bases.

In this sense, the results obtained lead to the idea that recuperating traditional habits, beliefs, and values can allow for sustainable economic development, which would inject value into natural wealth and its conservation.

## Figures and Tables

**Figure 1 ijerph-17-02283-f001:**
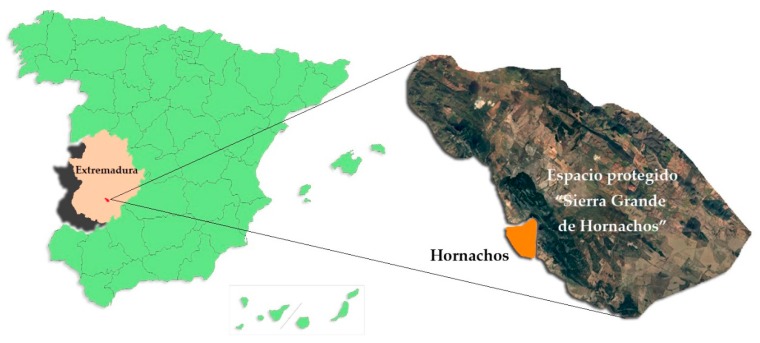
Location of the town of Hornachos and the protected area “Sierra Grande de Hornachos” in Extremadura, Spain.

**Figure 2 ijerph-17-02283-f002:**
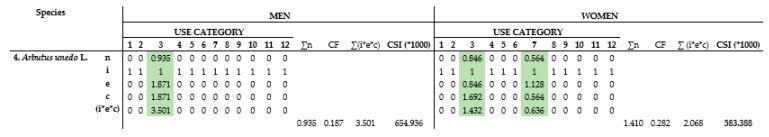
Excel spreadsheet model to introduce row data from paper cards.

**Figure 3 ijerph-17-02283-f003:**
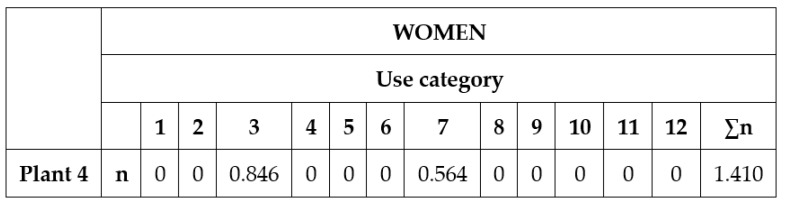
Example data from Table 1 (paper card) transcribed to a spreadsheet structure (first row of Figure 2).

**Figure 4 ijerph-17-02283-f004:**
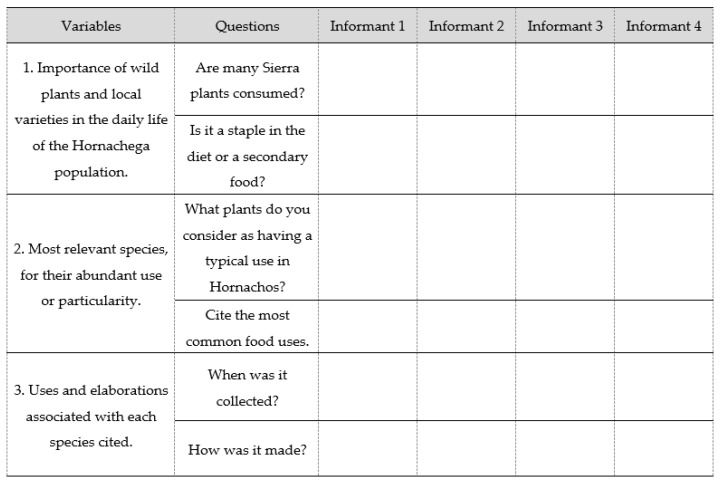
Spreadsheet model to enter the data extracted from interviews.

**Figure 5 ijerph-17-02283-f005:**
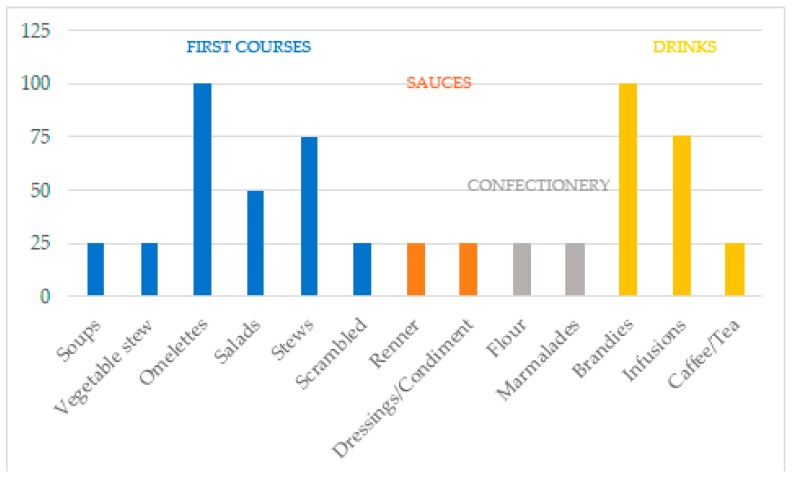
Culinary elaborations according to the percentage of interviewees that mentioned them.

**Table 1 ijerph-17-02283-t001:** Table design, with sample data.

Reference Number: Plant 4 (See List in Materials and Methods)			
N.º People that Consider it Useful Plan: 39. Sex: Women			
Category	N.º Persons Who Cite It in:	Select the Most Frequent Use (Only 1) *& Sing it as +F*	Select the Preferred Use (Only 1) *& Sing it as +P*
1. Vegetables	0		
2. Roots, bulbs, tubers, and rhizomes	0		
3. Sweet fruits	33	+F	
4. Dry and oleaginous fruits	0		
5. Cereals and pseudocereals	0		
6. Fat	0		
7. Alcoholic drinks	22		+P
8. Non-alcoholic drinks	0		
9. Condiments	0		
10. Sugars and sweeteners	0		
11. Candies and chewing	0		
12. Other food uses	0		

Source: Table made on the basis of [15].

**Table 2 ijerph-17-02283-t002:** Questionnaire.

Name and Surname of the Interviewee:			Date of Completion:	
Interviewer’s First and Last Names:			Time of Completion:	
Nº Variable	Issues	Species Cited		Other Annotations
**1**	Are many Sierra plants consumed? Is it a staple in the diet or a secondary food?			
**2**	What plants do you consider as having a typical use in Hornachos? Cite the most common food uses.			
**3**	When was it collected? How was it made?			

**Table 3 ijerph-17-02283-t003:** List of the 79 taxa studied ordered from highest to lowest value for CSIm + CSIw. In addition, the values of “n” and “CSI” are included. When CSIw > CSIm or CSIm > CSIw, this appears in bold. + = cultivated taxa.

Species	MEN		WOMEN		TOTAL
	∑n	CSIm*1000	∑n	CSIw*1000	(CSIm + CSIw)*1000
30. *Foeniculum vulgare* Mill.	2.710	**2539.371**	2.256	1292.726	**3832.097**
1. *Allium ampeloprasum* L.	1.935	1765.850	2.000	**2000.000**	**3765.850**
35. *Laurus nobilis* L.	1.000	1600.000	1.051	**1683.157**	**3283.157**
31. *Fragaria vesca* L.	1.000	1600.000	1.000	1600.000	**3200.000**
51. *Pinus pinea* L.	1.000	1600.000	1.000	1600.000	**3200.000**
59. *Rubus ulmifolius* Schott	1.065	852.056	1.718	**1508.090**	**2360.146**
46. *Origanum vulgare* subsp. *virens* (Hoffmanns. & Link) Bonnier & Layens	1.710	**1539.955**	0.872	499.462	**2039.417**
54. *Quercus rotundifolia* Lam.	1.000	800.000	1.026	**820.648**	**1620.648**
4. *Arbutus unedo* L.	0.935	**654.936**	1.410	583.388	**1238.324**
25. *Crataegus monogyna* Jacq.	1.097	**724.031**	0.872	469.772	**1193.803**
53. *Portulaca oleracea* L.	0.871	**528.562**	0.718	296.054	**824.616**
38. *Malva sylvestris* L.	0.968	**297.271**	0.949	290.666	**587.937**
42. *Mentha pulegium* L.	0.806	**419.590**	0.487	92.503	**512.093**
76. *Urtica urens* L.	0.774	**371.226**	0.282	14.872	**386.098**
58. *Rosmarinus officinalis* L.	0.742	**326.730**	0.051	0.108	**326.837**
26. *Cynara humulis* L.	0.323	26.854	0.872	**188.000**	**214.854**
62. *Scirpoides holoschoenus* (L.) Soják	0.161	3.357	0.615	**186.436**	**189.793**
50. *Phlomis lychnitis* L.	0.065	0.215	0.615	**186.436**	**186.651**
22. *Cichorium intybus* L.	0.419	**42.415**	0.282	6.305	**48.720**
21. *Chondrilla juncea* L.	0.419	**42.415**	0.231	3.217	**45.632**
64. *Silene vulgaris* (Moench) Garcke.	0.290	**19.576**	0.154	2.913	**22.489**
71. *Thymus mastichina* (L.) L.	0.290	**15.528**	0.179	3.422	**18.950**
56. *Reichardia intermedia* (Sch. Bip.) Samp.	0.194	**5.800**	0.103	0.863	**6.664**
24. *Cistus ladanifer* L.	0.129	**1.719**	0.051	0.108	**1.827**
13. *Calamintha nepeta* (L.) Savi	0.129	**1.719**	0.026	0.013	**1.732**
11. *Borago officinalis* L.	0.000	0.000	0.103	**0.863**	**0.863**
44. *Olea europaea* subsp. *europaea* var. *sylvestris* (Mill.) Lehr.	0.097	0.101	0.179	**0.708**	**0.809**
65. *Silybum marianum* (L.) Gaertn.	0.065	**0.215**	0.077	0.172	**0.387**
55. *Quercus suber* L.	0.097	**0.201**	0.000	0.000	**0.201**
3. *Andryala integrifolia* L.	0.032	0.027	0.051	**0.108**	**0.135**
34. *Hypericum perforatum* L.	0.000	0.000	0.051	**0.108**	**0.108**
68. *Sonchus oleraceus* L.	0.000	0.000	0.051	**0.108**	**0.108**
16. *Capsella bursa-pastoris* (L.) Medik	0.032	**0.027**	0.000	0.000	**0.027**
75. *Urginea maritima* (L.) Baker	0.032	**0.027**	0.000	0.000	**0.027**
43. *Myrtus communis* L.	0.000	0.000	0.026	**0.013**	**0.013**
48. *Papaver rhoeas* L.	0.000	0.000	0.026	**0.013**	**0.013**
2. *Allium neapolitanum* Cirillo	0.000	0.000	0.000	0.000	0.000
5. *Arisarum simorrhinum* Durieu	0.000	0.000	0.000	0.000	0.000
6. *Arum italicum* Mill.	0.000	0.000	0.000	0.000	0.000
7. *Arundo donax* L.	0.000	0.000	0.000	0.000	0.000
8. *Asphodelus albus* Mill.	0.000	0.000	0.000	0.000	0.000
9. *Bellis perennis* L.	0.000	0.000	0.000	0.000	0.000
10. *Bituminaria bituminosa* (L.) C.H. Stirt.	0.000	0.000	0.000	0.000	0.000
12. *Bryonia dioica* Jacq.	0.000	0.000	0.000	0.000	0.000
14. *Calendula arvensis* (Vaill.) L.	0.000	0.000	0.000	0.000	0.000
15. *Campanula rapunculus* L.	0.000	0.000	0.000	0.000	0.000
17. *Carthamus lanatus* L.	0.000	0.000	0.000	0.000	0.000
18. *Ceterach officinarum* Will.	0.000	0.000	0.000	0.000	0.000
19. *Chenopodium murale* L.	0.000	0.000	0.000	0.000	0.000
20. *Chenopodium* sp.	0.000	0.000	0.000	0.000	0.000
23. *Cistus albidus* L.	0.000	0.000	0.000	0.000	0.000
27. *Cytinus hypocistis* (L.) L.	0.000	0.000	0.000	0.000	0.000
28. *Digitalis thapsi* L.	0.000	0.000	0.000	0.000	0.000
29. *Dittrichia viscosa* (L.) Greuter	0.000	0.000	0.000	0.000	0.000
32. *Galactites tomentosus* Moench.	0.000	0.000	0.000	0.000	0.000
33. *Helichrysum stoechas* (L.) Moench	0.000	0.000	0.000	0.000	0.000
36. *Lonicera etrusca* Santi	0.000	0.000	0.000	0.000	0.000
37. *Lonicera implexa* Aiton	0.000	0.000	0.000	0.000	0.000
39. *Mantisalca salmantica* (L.) Briq. & Cavill.	0.000	0.000	0.000	0.000	0.000
40. *Marrubium vulgare* L.	0.000	0.000	0.000	0.000	0.000
41. *Medicago orbicularis* (L.) Bartal.	0.000	0.000	0.000	0.000	0.000
45. *Onobrychis humilis* (L.) G. López	0.000	0.000	0.000	0.000	0.000
47. *Oxalis corniculata* L.	0.000	0.000	0.000	0.000	0.000
49. *Phillyrea angustifolia* L.	0.000	0.000	0.000	0.000	0.000
52. *Pistacia lentiscus* L.	0.000	0.000	0.000	0.000	0.000
57. *Retama sphaerocarpa* (L.) Boiss.	0.000	0.000	0.000	0.000	0.000
60. *Ruscus aculeatus* L.	0.000	0.000	0.000	0.000	0.000
61. *Scandix pecten-veneris* L.	0.000	0.000	0.000	0.000	0.000
63. *Sedum brevifolium* DC.	0.000	0.000	0.000	0.000	0.000
66. *Smilax aspera* L.	0.000	0.000	0.000	0.000	0.000
67. *Solanum nigrum* L.	0.000	0.000	0.000	0.000	0.000
69. *Tamus communis* L.	0.000	0.000	0.000	0.000	0.000
70. *Thapsia villosa* L.	0.000	0.000	0.000	0.000	0.000
72. *Ulmus minor* Mill.	0.000	0.000	0.000	0.000	0.000
73. *Umbilicus gaditanus* Boiss.	0.000	0.000	0.000	0.000	0.000
74. *Umbilicus rupestris* (Salisb.) Dandy	0.000	0.000	0.000	0.000	0.000
77. *Veronica anagallis-aquatica* L.	0.000	0.000	0.000	0.000	0.000
78. *Viburnum tinus* L.	0.000	0.000	0.000	0.000	0.000
79. *Vinca difformis* Pourr.	0.000	0.000	0.000	0.000	0.000
**TOTAL**	**20.387**	**15,779.774**	**19.385**	**13,331.253**	**29,111.027**

**Table 4 ijerph-17-02283-t004:** List of taxa with the greatest importance for men ordered from highest to lowest value for CSIm*1000.

MEN	
Species	CSIm*1000
30. *Foeniculum vulgare* Mill.	**2539.371**
1. *Allium ampeloprasum* L.	**1765.850**
35. *Laurus nobilis* L.	**1600.000**
31. *Fragaria vesca* L.	**1600.000**
51. *Pinus pinea* L.	**1600.000**
46. *Origanum vulgare* subsp. *virens* (Hoffmanns. and Link) Bonnier and Layens	**1539.955**
59. *Rubus ulmifolius* Schott	**852.056**
54. *Quercus rotundifolia* Lam.	**800.000**
25. *Crataegus monogyna* Jacq.	**724.031**
4. *Arbutus unedo* L.	**654.936**
53. *Portulaca oleracea* L.	**528.562**
42. *Mentha pulegium* L.	**419.590**
76. *Urtica urens* L.	**371.226**
58. *Rosmarinus officinalis* L.	**326.730**
38. *Malva sylvestris* L.	**297.271**
22. *Cichorium intybus* L.	**42.415**
21. *Chondrilla juncea* L.	**42.415**
26. *Cynara humulis* L.	**26.854**
64. *Silene vulgaris* (Moench) Garcke.	**19.576**
71. *Thymus mastichina* (L.) L.	**15.528**
56. *Reichardia intermedia* (Sch. Bip.) Samp.	**5.800**
62. *Scirpoides holoschoenus* (L.) Soják	**3.357**
24. *Cistus ladanifer* L.	**1.719**
13. *Calamintha nepeta* (L.) Savi	**1.719**
50. *Phlomis lychnitis* L.	**0.215**
65. *Silybum marianum* (L.) Gaertn.	**0.215**
55. *Quercus suber* L.	**0.201**
44. *Olea europaea* subsp. *europaea* var. *sylvestris* (Mill.) Lehr.	**0.101**
3. *Andryala integrifolia* L.	**0.027**
16. *Capsella bursa-pastoris* (L.) Medik	**0.027**
75. *Urginea maritima* (L.) Baker	**0.027**

**Table 5 ijerph-17-02283-t005:** List of taxa with the greatest importance for women ordered from highest to lowest value for CSIw*1000.

WOMEN	
Species	CSIw*1000
1. *Allium ampeloprasum* L.	**2000.000**
35. *Laurus nobilis* L.	**1683.157**
31. *Fragaria vesca* L.	**1600.000**
51. *Pinus pinea* L.	**1600.000**
59. *Rubus ulmifolius* Schott	**1508.090**
30. *Foeniculum vulgare* Mill.	**1292.726**
54. *Quercus rotundifolia* Lam.	**820.648**
4. *Arbutus unedo* L.	**583.388**
46. *Origanum vulgare* subsp. *virens* (Hoffmanns. and Link) Bonnier and Layens	**499.462**
25. *Crataegus monogyna* Jacq.	**469.772**
53. *Portulaca oleracea* L.	**296.054**
38. *Malva sylvestris* L.	**290.666**
26. *Cynara humulis* L.	**188.000**
62. *Scirpoides holoschoenus* (L.) Soják	**186.436**
50. *Phlomis lychnitis* L.	**186.436**
42. *Mentha pulegium* L.	**92.503**
76. *Urtica urens* L.	**14.872**
22. *Cichorium intybus* L.	**6.305**
71. *Thymus mastichina* (L.) L.	**3.422**
21. *Chondrilla juncea* L.	**3.217**
64. *Silene vulgaris* (Moench) Garcke.	**2.913**
56. *Reichardia intermedia* (Sch. Bip.) Samp.	**0.863**
11. *Borago officinalis* L.	**0.863**
44. *Olea europaea* subsp. *europaea* var. *sylvestris* (Mill.) Lehr.	**0.708**
65. *Silybum marianum* (L.) Gaertn.	**0.172**
58. *Rosmarinus officinalis* L.	**0.108**
24. *Cistus ladanifer* L.	**0.108**
3. *Andryala integrifolia* L.	**0.108**
34. *Hypericum perforatum* L.	**0.108**
68. *Sonchus oleraceus* L.	**0.108**
13. *Calamintha nepeta* (L.) Savi	**0.013**
43. *Myrtus communis* L.	**0.013**
48. *Papaver rhoeas* L.	**0.013**

**Table 6 ijerph-17-02283-t006:** Species mentioned and the number of interviewees quoting them. Bold indicates one of the 79 species proposed in this study.

Vernacular Name	Scientific Name	Nº Informants Who Quote Her
Acelga bravía	*Beta* sp.	1
Ajo porro	*Allium ampeloprasum* L.	1
Huerbaluisa	*Aloysia citrodora* Gómez Ortega & Palau	1
**Madroño**	***Arbutus unedo* L.**	2
Espárrago	*Asparagus acutifolius L.*	4
Espárrago	*Asparagus albus* L.	4
Algarrobo	*Ceratonia siliqua* L.	1
Naranjo	*Citrus* × *sinensis* (L.) Osbeck	3
**Tilero/Guapero**	***Crataegus monogyna* Jacq.**	3
Membrillero	*Cydonia oblonga* Mill.	1
**Alcachofa borriquera**	***Cynara humulis* L.**	1
Higuera	*Ficus carica* L.	1
**Hinojo**	***Foeniculum vulgare* Mill.**	2
**Poleo**	***Mentha pulegium* L.**	1
**Orégano**	***Origanum vulgare* subsp. *virens* (Hoffmanns. and Link) Bonnier and Layens**	1
**Té del campo/Tila del campo**	***Phlomis lychnitis* L.**	3
Ciruelo	*Prunus domestica* L.	3
Albarillo	*Prunus armeniaca* L.	2
Guindero/guindillero	*Prunus cerasus* L.	2
Granado	*Punica granatum* L.	1
Pero de San Juan	*Malus domestica* (Borkh.) Borkh.	1
Coronillero	*Pyrus* sp.	2
Peral	*Pyrus communis* L.	1
**Encina**	***Quercus rotundifolia* Lam.**	2
**Alcornoque**	***Quercus suber* L.**	3
Berro	*Rorippa nasturtium-aquaticum* (L.) Hayek	3
**Zarzamora**	***Rubus ulmifolius* Schott**	1
Romaza	*Rumex pulcher* L.	3
Tagarnilla	*Scolymus hispanicus* L.	3
**Tomillo**	***Thymus mastichina* (L.) L.**	2
Tomillo	*Thymus zygis* Loefl. ex L.	2

**Table 7 ijerph-17-02283-t007:** Uses obtained in Hornachos contrasted with the data collected in the Spanish Inventory of Traditional Knowledge [20]. This includes uses that already appear in Extremadura (green), new uses in Extremadura (purple), and new uses in Spain (yellow). Hornachos rejected uses are marked with an “X” by their relative interpretation. In bold the highest number of uses obtained by one specie.

Nº Species with Uses	Species	1	2	3	4	5	6	7	8	9	10	11	12	Nº Uses	Subcategories of Use
1	*1. Allium ampeloprasum* L.													2	1, 2
2	*3. Andryala integrifolia* L.													1	1
3	*4. Arbutus unedo* L.													2	3, 7
4	*11. Borago officinalis* L.													1	1
5	*13. Calamintha nepeta* (L.) Savi													2	8, 9
6	*16. Capsella bursa-pastoris* (L.) Medik													1	1
7	*21. Chondrilla juncea* L.													3	1, 2, 8
8	*22. Cichorium intybus* L.													3	1, 2, 8
9	*24. Cistus ladanifer* L.													1	8
10	*25. Crataegus monogyna* Jacq.													3	3, 8, 11
11	***26. Cynara humulis* L.**													2	1, 11
12	*30. Foeniculum vulgare* Mill.													5	1, 7, 8, 9, 11
13	*31. Fragaria vesca* L.													1	3
14	*34. Hypericum perforatum* L.													1	8
15	*35. Laurus nobilis* L.													2	8, 9
16	*38. Malva sylvestris* L.													2	8, 11
17	*42. Mentha pulegium* L.													1	8
18	*43. Myrtus communis* L.													1	3
19	*44. Olea europaea* subsp. *europaea* var. *sylvestris* (Mill.) Lehr.													3	4, 6, 8
20	*46. Origanum vulgare* subsp. *virens* (Hoffmanns. and Link) Bonnier and Layens													2	8, 9
21	*48. Papaver rhoeas* L.													1	8
22	*50. Phlomis lychnitis* L.													2	7, 8
23	*51. Pinus pinea* L.													1	4
24	*53. Portulaca oleracea* L.													1	1
25	*54. Quercus rotundifolia* Lam.													2	4, 8
26	***55. Quercus suber* L.**													2	4, 8
27	*56. Reichardia intermedia* (Sch. Bip.) Samp.													1	1
28	*58. Rosmarinus officinalis* L.													1	9
29	*59. Rubus ulmifolius* Schott													**5**	1, 3, 7, 8, 11
30	*62. Scirpoides holoschoenus* (L.) Soják													1	11
31	*64. Silene vulgaris* (Moench) Garcke.													1	1
32	*65. Silybum marianum* (L.) Gaertn.													2	1, 8
33	*71. Thymus mastichina* (L.) L.													2	8, 9
34	***75. Urginea maritima* (L.) Baker**													1	2
35	*76. Urtica urens* L.								X					2	1, 8

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
