# Peer review of "Food Identities, Biocultural Knowledge and Gender Differences in the Protected Area “Sierra Grande de Hornachos” (Extremadura, Spain)"

_ijerph, 2020, doi:10.3390/ijerph17072283_

Round 1
Reviewer 1 Report
The title "Food Identities, Biocultural Knowledge and Sustainable Agri-Food Systems: An Agro-Ecological Approach to Development in the Protected Area “Sierra Grande de Hornachos” (Extremadura, Spain)" describes the article.
The keywords are appropriate
Page 2, last paragraph: where is "described by [10], who speaks of an implementation phase..." should be "described by Rodríguez Guerra and co-workers [10], who speaks of an implementation phase..."
Page 4, Table 1 caption: where is "Table desing, with sample data." should be "Table design, with sample data."
Table 1 must be on the same page, not divided.
Page 9: The number of the equation CSI = CF*[Σ(i*e*c)i] = [Σn/H]*[Σ(i*e*c)i], is missing.
Page 10: Table 3 doesn't fit on the same page, so, the caption in the followed page should be:
Table 3. List of the 79 taxa studied ordered from highest to lowest value for CSIm + CSIw. In addition, the values of ‘n’ and ‘CSI’ are included. When CSIw >CSIm or CSIm > CSIw appears in bold. + = cultivated taxa. [Continnued]
Page 12: Table 5.must be on the same page, not divided.
Page 13: Table 6.must be on the same page, not divided.
Page 16: Table 7 must be on the same page, not divided.
Author Response
Reviewer #1
The title "Food Identities, Biocultural Knowledge and Sustainable Agri-Food Systems: An Agro-Ecological Approach to Development in the Protected Area “Sierra Grande de Hornachos” (Extremadura, Spain)" describes the article. The keywords are appropriate
Thank you very much for the appreciation of our work and the time spent on its carefully revision. We have assumed the changes you have signed and we have made the following asked modifications:
- Page 2, last paragraph: where is "described by [10], who speaks of an implementation phase..." should be "described by Rodríguez Guerra and co-workers [16], who speaks of an implementation phase..."
Which has been changed as required, see line 112.
- Page 4, Table 1 caption: where is "Table desing, with sample data." should be "Table design, with sample data."
The word design has been properly written, see line 228.
- Table 1 must be on the same page, not divided.
We agree with this correction and we have made the corresponding changes. We hope that the MDPI Editing Service will be able to help in this topic. For this reason we have used the MDPI Layout Services, which certificate was uploaded to the platform. We are sure this mistake will be corrected in the final version of the corresponding accepted manuscript.
- Page 9: The number of the equation CSI = CF*[Σ(i*e*c)i] = [Σn/H]*[Σ(i*e*c)i], is missing.
We have added now a (1) quotation that was lack before, see line 327.
- Page 10: Table 3 doesn't fit on the same page, so, the caption in the followed page should be: Table 3. List of the 79 taxa studied ordered from highest to lowest value for CSIm + CSIw. In addition, the values of ‘n’ and ‘CSI’ are included. When CSIw >CSIm or CSIm > CSIw appears in bold. + = cultivated taxa. [Continnued]
We have made the required change, as it can be seen in lines 370-372, 373-375, 377-379.
- Page 12: Table 5.must be on the same page, not divided.
We agree with this correction and we have assumed it in the same way as in Table 1, 6 and 7.
- Page 13: Table 6.must be on the same page, not divided.
We agree with this correction and we have assumed it in the same way as in Table 1, 5 and 7.
- Page 16: Table 7 must be on the same page, not divided.
We agree with this correction and we have assumed it in the same way as in Table 1, 5 and 6.
Reviewer 2 Report
This is an interesting topic because addresses the construction of food identity and the importance of traditional local knowledge of rural people. The paper contains significant information adequate to justify publication. However, I am not sure if the content of the paper really fits the scope of the IJERPH journal, according to https://www.mdpi.com/journal/ijerph/about
The paper brings a strong debate about food identities and consecuently biocultural knowlededge, which is verified with data obtained, but the methodology applied is not directly linked to study the sustainability of agrifood systems. It is possible to defend the relationship between these elements theoretically, but sustainability would have to be studied not only theoretically. I did not find data in this paper to justify the link between food identities and sustainable agri-food systems. Then, the tittle is inadequate.
The introduction section addresses, maily, the loss of food culture, cultural wealth and food identity. There is not information about sustainability.
The introduction section also should address the issues you point out in the results discussion section, such as: plant species, uses and gender differences.
With the elements you have, the paper should end with a well-constructed conclusion. For example, which are the main workshop results?
Something that caught my attention is the lack of scientific references to support the introduction and discussions in this study.
Please, escape unnecessary quotation marks: “particularity”, “food culture”, “wisdom”, “intangible heritage”, “patrimonialization”; etc.
There are some typing mistakes
Author Response
This is an interesting topic because addresses the construction of food identity and the importance of traditional local knowledge of rural people. The paper contains significant information adequate to justify publication. However, I am not sure if the content of the paper really fits the scope of the IJERPH journal, according to https://www.mdpi.com/journal/ijerph/about
First of all, thank you for the assessment you make of our work . Our submission to the special issue "Interdisciplinary Approach to Improve AgriFood Safety and Quality" of the IJERPH journal was promoted by the formal invitation we receive for participating in that special issue, due to our previous experience in the topic. We honestly believe that the conceptual and methodological framework that we have developed fits into the framework defined in the presentation of the Special Issue, as it can be proved in these sentences:
“Since the use of an interdisciplinary approach is essential, this Special Issue requires documents which demonstrate the important role that the integrated and multidisciplinary perspective can play in relation to food safety problems with a vision of Food Hygiene, Public Health, Environmental Sciences, Food Chemistry, Economics, and/or others themes. The aim of this Special Issue is to collect innovative papers deriving from different and complementary expertises concerning “AgriFood Safety and Quality.”
The paper brings a strong debate about food identities and consecuently biocultural knowldedge, which is verified with data obtained, but the methodology applied is not directly linked to study the sustainability of agrifood systems. It is possible to defend the relationship between these elements theoretically, but sustainability would have to be studied not only theoretically.
We totally agree, and for that reason we have included strictly quantitative methodologies that even have statistical treatment, semi-qualitative methodologies and qualitative analyzes throughout the work.
I did not find data in this paper to justify the link between food identities and sustainable agri-food systems.
Our approach on sustainability of agri-food systems has been built up from the relocalisation paradigm, and most of the papers from the biological areas fit in much more in the sustainanble development paradigm, which is focused on environment and agriculture. Sociologists more frequenty are situated under the scope of the relocalization one. To clarify this starting point we have added some explicative lines (50-53; 57-60).
The tittle is inadequate.
We have changed the title to “Food Identities, Biocultural Knowledge and gender differences in the Protected Area “Sierra Grande de Hornachos” (Extremadura, Spain).“
The introduction section addresses, maily, the loss of food culture, cultural wealth and food identity. There is not information about sustainability.
As we explained in the abovementioned paragraphs, we have added these sentences:
Line 50: Agri-food systems are the processes and infrastructure involved in feeding a population. The sustainability of agri-food systems can be approached by two paradigms the sustainable development (focused on agriculture and the environment) and the relocalisation paradigm. The latter very used by sociologists and mainly based on alternative food systems.
Line 58: An approach dealing with agriculture, food and environment should consider the diversities of potentialies and initiatives in each specific territory. The first steps including an ethnographic assessment allowing for the identification of traditional knowledge.
The introduction section also should address the issues you point out in the results discussion section, such as: plant species, uses and gender differences.
We agree with this suggestion and we have made the following changes
- Plant species: lines 97-100.
- Uses: lines 103-104.
- Gender differences: lines 105-110.
With the elements you have, the paper should end with a well-constructed conclusion. For example, which are the main workshop results?
Yes, you are right. Conclusions will be more solid if we include the results obtained from the quantitative experimental part of the work. We have written a new paragraph of conclusions in lines 524-528.
Something that caught my attention is the lack of scientific references to support the introduction and discussions in this study.
We have taken in consideration this observation, and we have gone through the text, including additional literature we had already considered but we had not included in order to make the paper not so long. But finally we have added them. You can check that now we have 31 references instead of 26 of the original submitted version of the manuscript.
Please, escape unnecessary quotation marks: “particularity”, “food culture”, “wisdom”, “intangible heritage”, “patrimonialization”; etc. There are some typing mistakes
Yes you are right. We have deleted them in lines 17, 45, 54, 61, 71, 72, 93, 155 436. We have asked the MDPI Editing Service to do their best in order to improve the last version of the manuscript . (Uploaded Certificate)
Round 2
Reviewer 2 Report
The authors have made changes based on comments from the previous review. I agree with the publication of this article.